# Recently Updated Role of Chitinase 3-like 1 on Various Cell Types as a Major Influencer of Chronic Inflammation

**DOI:** 10.3390/cells13080678

**Published:** 2024-04-14

**Authors:** Emiko Mizoguchi, Takayuki Sadanaga, Linda Nanni, Siyuan Wang, Atsushi Mizoguchi

**Affiliations:** 1Department of Immunology, Kurume University School of Medicine, Kurume 830-0011, Japan; taka.sadanaga@gmail.com (T.S.); wang0627yan@gmail.com (S.W.); mizoguchi_atsushi@med.kurume-u.ac.jp (A.M.); 2Department of Molecular Microbiology and Immunology, Brown University Alpert Medical School, Providence, RI 02912, USA; 3Catholic University of the Sacred Heart, 00168 Rome, Italy; lindasismup@gmail.com

**Keywords:** chitinase 3-like 1, chronic inflammation, epithelial cells, animal models, host–microbial interactions

## Abstract

Chitinase 3-like 1 (also known as CHI3L1 or YKL-40) is a mammalian chitinase that has no enzymatic activity, but has the ability to bind to chitin, the polymer of N-acetylglucosamine (GlcNAc). Chitin is a component of fungi, crustaceans, arthropods including insects and mites, and parasites, but it is completely absent from mammals, including humans and mice. In general, chitin-containing organisms produce mammalian chitinases, such as CHI3L1, to protect the body from exogenous pathogens as well as hostile environments, and it was thought that it had a similar effect in mammals. However, recent studies have revealed that CHI3L1 plays a pathophysiological role by inducing anti-apoptotic activity in epithelial cells and macrophages. Under chronic inflammatory conditions such as inflammatory bowel disease and chronic obstructive pulmonary disease, many groups already confirmed that the expression of CHI3L1 is significantly induced on the apical side of epithelial cells, and activates many downstream pathways involved in inflammation and carcinogenesis. In this review article, we summarize the expression of CHI3L1 under chronic inflammatory conditions in various disorders and discuss the potential roles of CHI3L1 in those disorders on various cell types.

## 1. Introduction

Inflammatory Bowel Disease (IBD) includes two major diseases, Ulcerative Colitis (UC) and Crohn’s Disease (CD), both of which are diseases associated with chronic inflammation that are difficult to cure even if they go into remission. In most cases, patients must live with these chronic diseases for the rest of their lives [1,2,3]. Some mammalian chitinases, of which expression is mainly induced in cells involved in innate immunity such as epithelial cells and macrophages under chronic inflammation, including IBD, are also one of the factors that play a central role in host–microbial interactions and dysbiosis in animal models and humans [4,5,6,7,8,9]. Our group reported the involvement of mammalian chitinases in host–microbial interactions [7,8,9]. 

Chitinases produced by mammals and bacteria are glycosidases that break down the glycosidic bonds of chitin, a polymer of N-acetyl glucosamine. Chitin is a major component of the exoskeleton and cell walls of various organisms, including arthropods, nematodes, and fungi. However, its existence in vertebrates (including mice and humans) has not been confirmed. Among the 131 types of Glycoside Hydrolases (GH), chitinases are broadly classified into the families of GH18 and GH19 based on differences in their amino acid sequences, three-dimensional structures, and catalytic actions [10]. The GH18 chitinase family is divided into two types: authentic (true) chitinases, which have enzymatic activity, and Chitinase-Like Proteins (CLP), which do not have enzymatic activities. The former includes Chitotriosidase (chitinase 1) and Acidic mammalian chitinase (AMCase), and the latter includes Chitinase 3-like 1 (CHI3L1 or YKL-40) [10]. It has been suggested that CLPs act as endogenous lectins that recognize specific sugar chains, such as chitin and chito-oligosaccharide, and regulate cell adhesion, migration, differentiation, and proliferation [11]. True chitinases and CLPs share structural similarities, but they have different catalytic activities due to differences in their amino acid structures. CHI3L1 completely loses enzymatic activity due to the replacement of the glutamic acid residue with leucine residue in the chitin-binding site in its multidomain structure, but it remains a sufficient affinity for chitin [12,13]. Although many aspects of the in vivo functions of mammalian chitinases remain unclear, they are highly involved in immune responses in various organs and tissues. In particular, CHI3L1 is correlated with certain pathological conditions, including tissue damage responses and acute/chronic inflammation, and is likely to influence both innate and adaptive immunity [14]. 

Epithelial cells, in particular colonic epithelial cells (CECs), form a barrier mechanism as the first line of defense to avoid immune responses from external stimuli. In addition to absorbing nutrients and water, CECs protect intestinal tissues from many intestinal bacteria and form a two-layered mucin layer on the outside. The mucin layers, together with CECs, form a strong barrier between intestinal bacteria and the lamina propria, and greatly contribute to the maintenance of homeostasis in the intestinal tract. However, once this mucosal layer thins or is lost, an imbalance between the host and intestinal flora—so-called dysbiosis—occurs and results in inflammation of the intestinal tract [4,15]. CHI3L1 is highly involved in the host–microbial interactions by regulating the interactions between CECs and potentially pathogenic bacteria in the gut [7,9].

In this review article, we discuss the potential role of CHI3L1 expressed on various types of cells during inflammatory conditions in mice and humans. 

## 2. Potential Biological Roles of CHI3L1 

In 2004, Elias et al. reported the exciting fact of a formation of crystalline structures in lung tissues of a murine asthma model, and identified that the crystals were AMCase, one of the true mammalian chitinases [16]. In this report the authors discovered that AMCase is induced by Interleukin (IL)-13-mediated T helper-2 (Th2)-specific inflammatory signaling pathway in lung epithelial cells and macrophages [16]. Later studies revealed that not only AMCase but also CHI3L1 levels in serum as well as lung tissues were significantly upregulated in a joint cohort study from the USA and France [17] suggesting that serum CHI3L1 produced by various cell types becomes a good parameter for inflammatory conditions. A large number of reports have been published regarding the association between CHI3L1 and inflammatory conditions in the past nearly three decades [18,19,20,21,22]. Over the past five years, the role of CHI3L1 in various types of inflammation has been rapidly elucidated, as shown in Table 1 [23,24,25,26,27,28,29,30,31,32,33,34,35,36,37,38,39,40,41,42,43,44,45,46,47,48,49,50,51,52,53,54,55,56,57,58,59,60,61,62,63,64,65,66,67,68,69,70,71]. We also have summarized the potential roles of CHI3L1 in the four major diseases in Section 2.1 of this review. 

### 2.1. Increased Levels of CHI3L1 under the Major Inflammatory Disorders

#### 2.1.1. Inflammatory Bowel Disease (IBD)

The serum as well as tissue levels of CHI3L1 are significantly elevated in patients with IBD including UC and CD, and the elevation of serum CHI3L1 is primarily associated with the severity, the extent of inflammation, and the existence of complications such as arthritis [7,18,19,20,21]. Interestingly, serum CHI3L1 levels become high in CD patients with fibrosis, which appears in more severe cases, suggesting that CHI3L1 is a possible inflammatory biomarker in IBD [22]. Our group previously reported that the CHI3L1 mRNA expression levels were increased in active UC and involved the region of CD compared with inactive UC, the uninvolved region of CD, and normal individuals [7]. The apical sides of CECs seem to produce CHI3L1 protein, mainly in the active CD patients’ regions [7]. Since the expression pattern of CHI3L1 and bacterial biofilm formation are almost identical, it is easy to imagine the interaction between CHI3L1 expression and intestinal bacteria, in particular, potentially pathogenic bacteria. In the active phase of IBD, CHI3L1 is continuously secreted as a 40 kDa protein from CECs and macrophages into the intestinal lumens, and therefore, it is reasonable that not only serum but also fecal CHI3L1 seems to be a reliable biomarker for predicting the severity and activity of IBD [72,73]. 

Interestingly, fecal CHI3L1 expression levels were almost undetectable in healthy individuals and a non-significant step-wise increase in IBD patients under the remission phase (CRP < 0.1), but the levels were significantly upregulated in IBD patients with dysplasia/adenocarcinoma compared with other adenoma or sporadic colorectal cancer patients, suggesting that fecal CHI3L1 levels might be a non-invasive and reliable biomarker for IBD-associated malignant changes of CECs under the remission phase of IBD [74]. 

#### 2.1.2. Multiple Sclerosis (MS)

MS is an inflammatory demyelinating disease of the central nervous system and is characterized by multiple temporal and spatial occurrences. It is an intractable autoimmune disease that takes a chronic course and causes inflammation in the brain, spinal cord, and optic nerves, damaging nerve tissue. In 2010, Comabella et al. identified that CHI3L1 seems to be a prognostic biomarker for conversion to MS and development of disability utilizing the previously collected cerebrospinal fluid samples from MS patients by a mass spectrometry-based proteomic approach validating with ELISA (enzyme-linked immunosorbent assay) [75]. Positively associated elevation of CHI3L1 in MS samples was followed/reviewed by many other groups so far [27,28,29,30,31,32,33,34,35,76,77,78]. 

Of note, CHI3L1 expression in astrocytes is positively associated with increased expression of representative proinflammatory cytokines, IL-1 and IL-6. IL-1 and IL-6 or the IL-6 family cytokine, Oncostain M, synergistically upregulate CHI3L1 expression, which employs STAT3 and NF-κB to the promoter region of CHI3L1 [76]. Our group also previously proved that CHI3L1 and IL-6 synergistically activate STAT3 signaling pathways in intestinal epithelial cells in an NF-κB and MAPK-dependent manner [79], so it is extremely interesting that the same thing happens in brain tissue as well. In addition, dysregulated productions of TNF, another representative pro-inflammatory cytokine, and soluble TNF receptors type I and type II protein levels in CSF are associated with specific clinical profiles, and are useful to identify at a very early stage in MS patients, which is very useful for the prediction of the MS disease outcome [32]. Overall, CHI3L1 seems to be regulated by the signaling pathways of pro-inflammatory cytokines and their receptors in MS as well as IBD [7,32,76,79,80]. 

#### 2.1.3. Alzheimer’s Disease (AD) 

AD, a frequently occurring and debilitating disorder of the central nervous system (CNS), is classically viewed as a progressive neurodegenerative disorder resulting in intellectual impairment, memory loss, and spatial disorientation [81]. The hallmarks of AD pathology are the deposition of amyloid beta (Aβ) containing plaques and neurofibrillary tangles composed of hyperphosphorylated tau protein in the brain [82]. In the longitudinal early-onset AD study (LEADS), the results showed that cerebrospinal fluid (CSF) biomarkers were correlated with each other including CHI3L1, and CSF CHI3L1 was associated with cognition and astrocytic changes during the early onset of AD [41,83]. The CHI3L1 levels in CSF, also used as an astrocyte biomarker, increased very early in AD progression and mediated Aβ-induced tau phosphorylation and tau-induced neuronal injury. One study observed that CSF CHI3L1 levels were associated with tau pathology and the over-secreted CHI3L1 from astrocytes related to the accumulation of tau tangles in the living AD brains. These results suggest that CHI3L1 is an important mediator of key pathogenic events in the AD pathogenic cascade and contributes to AD progression [84,85]. In cells and mice, the deletion of CHI3L1 alters the responses of glial inflammation, promotes microglial Aβ and astrocyte phagocytosis, and decreases amyloid plaque deposition, but glial activation and neuroinflammation may be dependent on context because the deletion of CHI3L1 could be neuro-protective in AD, but destructive in acute inflammation. Thus, the upregulation of CHI3L1 suppresses microglial Aβ and astrocyte phagocytosis and accelerated amyloid plaque formation, which contributes to the progression of AD [86]. The increase of CHI3L1 was also associated with cognitive dysfunction, and CHI3L1 plays a significant role in white matter neuroinflammation associated with cognitive decline in AD patients, which suggests that white matter CHI3L1 relates to cognitive impairment in the early onset of AD [39]. Recent studies confirmed that DNA variants in CHI3L1 could be associated with increased neuronal injury and inflammation, and CSF levels of CHI3L1 could lead to an increased risk of AD. Also, the CHI3L1 expression in both blood and CSF is positively associated with variants in CHI3L1 [36,87]. The suppression of CHI3L1 DNA variants may contribute to lower levels of blood and CSF CHI3L1, which reduces the risk of AD development.

In conclusion, upregulation of CHI3L1 both in blood and CSF can contribute to the progression of AD, and the implications of anti-CHI3L1 therapies may enhance treatment responses in future clinical trials.

#### 2.1.4. Asthma, Chronic Obstructive Pulmonary Disease (COPD) 

Recent studies have demonstrated that the concentration of serum CHI3L1, which relates to the severity of the disease, is upregulated in patients with COPD. The elevation of serum CHI3L1 may contribute to tissue inflammation and remodeling by activating alveolar macrophages, which are both the target and the source of CHI3L1 [11,45,48]. Similarly, circulating CHI3L1 levels were also elevated in asthma patients compared with healthy controls and positively correlated with the severity of asthma [17,88].

It also has been demonstrated that a promoter −131C→G SNP (single nucleotide polymorphism) in CHI3L1 is associated with increased serum CHI3L1 levels and the severity of asthma [89]. A novel intronic SNP, rs12141494, alters airway CHI3L1 expression to contribute to the severity of asthma and airway remodeling. Although this SNP, unlike promoter SNP rs4950928 (−131C > G), was associated with CHI3L1 expression in the sputum, there was no association with asthma severity. Furthermore, the A allele of rs4950928 was associated with higher serum CHI3L1 levels and severer asthma after the control of risk genotype (CC). The A allele of rs12141494 has significantly higher CHI3L1 sputum levels compared to the G allele [90,91,92]. These results suggest that CHI3L1 is an intermediate phenotype for asthma susceptibility, and DNA variants in CHI3L1 play important roles in the progression of severe asthma and airway remodeling. Thus, the inhibition of CHI3L1 DNA variants could contribute to the lower production of circulating CHI3L1 and decrease the risk of asthma and airway remodeling.

A prospective cohort design found that serum CHI3L1 level relates to the increase of the risks from moderate to severe asthma exacerbations and can be a predictor of moderate to severe asthma exacerbation. Furthermore, CHI3L1 is a signature of non-type 2 inflammation for NEA (non-eosinophilic asthma) patients and increased serum CHI3L1 levels are associated with NEA phenotypes [48]. Murine studies have found that CHI3L1 was induced by a high-fat diet and Th2 inflammation (such as asthma) and contributes to the genesis of obesity. Serum CHI3L1 was also associated with persistent asthma in obese asthma patients. However, sputum CHI3L1 expression was associated with only truncal obesity in humans [92]. Thus, the inhibition of CHI3L1 or CHI3L1 pathways could provide potential therapeutic treatments for obesity-related asthma.

### 2.2. CHI3L1 Expression in Various Cell Types

The CHI3L1 expression was first identified in human synovial cells, articular cartilage chondrocytes, and osteosarcoma cells [93]. Johansen et al. identified that the soluble form of CHI3L1 levels in serum and synovial fluid were significantly higher in patients with joint disease as compared to normal adults [93]. Their continuous studies of CHI3L1 in various inflammatory disorders as well as malignant diseases revealed that CHI3L1 was produced by inflammatory cells and cancer cells by regulating cell proliferation, differentiation, and extracellular tissue remodeling [94]. Our group first reported that the apical side of CECs, as well as lamina propria cells, strongly express CHI3L1 in several murine colitis models and IBD patients but is completely absent in normal controls/individuals, suggesting that CHI3L1 is an inducible molecule under inflammatory conditions in the colon and plays a pathogenic role in colitis [7]. We also reported that CHI3L1 on CECs is further upregulated during the processes of colitis-associated cancer [74]. The CHI3L1 expression in HCT116 human CEC line is observed mainly in peri-nuclear and cytoplasm regions (Figure 1A), although the location was mainly restricted in the nucleus with reduced expression of CD44 after stimulating with human TNFα recombinant protein for 48 h (Figure 1B). This restricted trans-nuclear localization was inhibited by combinational treatment with pan-chitinase inhibitors (caffeine and pentoxifylline) (Figure 1C), suggesting that this nuclear localization of CHI3L1 after TNFα stimulation seems to be specifically controlled by chitinase activity. In the future, we plan to study the mechanism by which CHI3L1 translocates into the nucleus and whether there are changes in its biological function after the translocation.

It has been reported that CHI3L1 expression is highly upregulated with cancer-infiltrating macrophages, so-called TAM (tumor-associated macrophages) [95]. CHI3L1 specifically promotes macrophage recruitment and tumor angiogenesis in colon cancer [96]. Alternatively activated macrophages (M2 Mφ) but not classically activated macrophages (M1 Mφ) generally express mammalian chitinases, including CHI3L1, CHI3L3 (YM1), and CHI3L4 (YM2) under the activation of MAPK pathway [97]. 

Brain tumors such as glioma highly express CHI3L1 by interacting with CD44 on the surface of glioma stem cells (GSCs) that result in activating the Akt and β-catenin signaling cascades [98]. The activation, in turn, upregulates CD44 expression in a pro-mesenchymal feedback loop [98]. The CHI3L1 expression seems to alter the state of GSCs to support tumor growth and regulate cellular plasticity, leading to a targetable vulnerability to glioblastoma [99]. This result suggests that a blockade of CHI3L1 by specific antibody may serve as one of the helpful therapeutic strategies for glioblastoma.

### 2.3. CHI3L1-Associated Chronic Inflammation in Animal Models 

Previous research has revealed that CHI3L1 is involved in various inflammatory diseases, and plays a significant role in inflammatory conditions. The involvement of CHI3L1 across such a broad spectrum of diseases highlights its biological significance in understanding the pathogenesis of chronic inflammation. Using animal models has allowed us to assess the significance of CHI3L1 in numerous chronic inflammations listed in Table 2 [59,74,79,92,100,101,102,103,104,105,106,107,108,109,110,111,112,113,114,115,116]. Intriguingly, beyond the chronic inflammations delineated in Section 2.1, emerging evidence indicates the modulation of cardiovascular diseases, liver injuries, kidney diseases, systemic musculoskeletal disorders, and obesity by CHI3L1. 

Tsantilas et al. employed ApoE knockout (KO) mice subject to a high-fat diet to investigate the atherosclerosis plaque rupture mechanism, revealing CHI3L1 plays a regulatory role in plaque size and vulnerability [110]. These findings elucidated that CHI3L1 expression is induced by pro-inflammatory cytokines, such as IL-1β, and IL-6, originating from smooth muscle cells within the plaque. Moreover, CHI3L1 KO resulted in smooth muscle cell apoptosis and the formation of larger, unstable, or ruptured plaques [111]. Notably, CHI3L1 induction extends beyond pro-inflammatory cytokines, as demonstrated by Lee et al., who unveiled that CHI3L1 facilitates oxidative stress and chronic inflammation in the alcohol liver injury rat model [112]. CHI3L1 was found to upregulate the expression of iNOS (inducible nitric oxide synthase), COX-2 (cyclooxygenase-2), TNFα, IL-1β, and chemokines such as MIP-1α and MIP-1β in the liver. Additionally, CHI3L1 KO rats in the alcohol injury model exhibited suppressed levels of ICAM-1, suggesting the involvement of CHI3L1 in neutrophil-mediated inflammation in the liver [113,114]. CHI3L1 is widely acknowledged as a potent inducer of Th2-type immune responses [115,116]. However, numerous studies suggest that CHI3L1 also exerts a robust induction of cytokines, particularly in certain chronic inflammatory contexts, by modulating IL-6 production or STAT3-mediated signaling activation [79,102].

Collectively, these findings underline the collaborative roles of CHI3L1 with various cell types in modulating inflammatory responses and establishing feedback loops across diverse chronic inflammatory contexts. Therefore, the animal models of chronic inflammation have been valuable in elucidating the significant mechanistic impact of CHI3L1 and have advanced our development of novel therapeutic strategies.

### 2.4. CHI3L1-Associated Chronic Inflammation in Human 

Much evidence has highlighted the significance of CHI3L1 in various human inflammatory conditions, as described in Table 1. A wide range of studies have elucidated a robust correlation between CHI3L1 expression levels and the severity as well as prognostic outcomes of diseases, including asthma, atopic dermatitis, and interstitial lung disease [90,117,118]. This collective body of research underlines the pivotal role of CHI3L1 in the pathophysiology of inflammatory disorders, providing valuable insights into its potential as a diagnostic and prognostic biomarker.

The pathogenesis of CHI3L1 in chronic inflammation exhibits disease-specific variations (Figure 2). In pulmonary disorders such as asthma or Hermansky-Pudlak syndrome (HPS), a predominant Th2 immune response orchestrates chronic inflammatory processes [106,108]. In asthma, this immune activation is initiated as dendritic cells engage with allergens and pathogens presented on epithelial cells, subsequently inducing Th2 cell differentiation. Moreover, damaged epithelial cells contribute to Th2 cell activation by secreting IL-25 and IL-33, which also stimulate innate lymphoid cell type 2 (ILC2) [119]. The resultant Th2-mediated inflammation not only manifests as eosinophilia and mast cell degranulation but also fosters the polarization of M0 macrophages towards the M2 phenotype. Notably, IL-13 emerges as a critical mediator in inducing CHI3L1 expression in both epithelial cells and M2 macrophages. Consequently, elevated CHI3L1 expression in epithelial cells exacerbates pathogen infiltration, establishes a feedback loop, and perpetuates chronic inflammation [106] (Figure 2). 

In contrast to pulmonary conditions, IBD presents a distinct chronic inflammatory profile modulated by CHI3L1. Our previous investigations revealed a notable increase in CHI3L1 expression within CECs during the course of chronic inflammation [7,9,74]. Additionally, our research demonstrated the pivotal role of CHI3L1 in facilitating potentially pathogenic bacterial adhesion to CECs [79]. In this context, bacterial interaction with Toll-like receptor 4 (TLR4) on epithelial cells initiates the production of pro-inflammatory cytokines, including IL-1β, IL-6, and TNF-α, via MyD88, TRIF, and MAPK signaling pathways [120]. These cytokines further stimulate the upregulation of CHI3L1 expression in epithelial cells. Moreover, CHI3L1 exerts direct effects on SW480, a human colorectal cancer cell line, by activating the NF-κB and MAPK pathways, consequently upregulating the expression of pro-inflammatory cytokines/chemokines such as TNFα, IL-8, and CCL2 (C-C motif chemokine ligand 2). Activation of these signaling promotes macrophage recruitment and enhances angiogenesis within the tumor microenvironment, thereby fostering tumor growth [96,121].

CHI3L1 has been identified as a potent modulator of immune responses, particularly through its stimulation of macrophage and neutrophil activity, as well as its influence on immune checkpoints, thereby establishing a conducive environment for tumor growth. Notably, CHI3L1 has been shown to drive the polarization of macrophages towards the M2 phenotype, commonly referred to as TAMs (tumor-associated macrophages), within the tumor microenvironment [121,122,123]. TAMs are recognized for their pivotal roles in various aspects of tumor progression, including tumor development, neo-angiogenesis modulation, immune suppression, and metastasis [98]. Moreover, recent findings have underscored the role of CHI3L1 in inducing NETosis, thereby facilitating the exclusion of T cells and promoting the establishment of triple-negative breast cancer tumors [124]. These observations collectively highlight the multifaceted involvement of CHI3L1 in shaping the inflammatory tumor microenvironment and influencing cancer progression.

## 3. CHI3L1-Mediated Host–Microbial Interactions

### 3.1. CHI3L1 as an Enhancer of Bacterial Adhesion and Invasion on CECs

The interplay between the intestinal microbiome and the gastrointestinal (GI) tract has been extensively presented as bidirectional [125]: reciprocal signaling occurs between the bacterial flora and the mucosal immune system, thus modulating gut homeostasis. It has been predicted that the altered expression of specific receptor(s) on host intestinal epithelial cells might enhance the interaction with bacterial components under inflammatory conditions [126]. Among these molecules, CHI3L1 has been targeted as a potential enhancer of bacterial adhesion and invasion on/into CECs [7,127].

Microbial chitinases, which are generally associated with chitinolytic activity for nutritious purposes, have recently been linked to bacterial virulence. Although mammals do not synthesize chitin, *Listeria Pneumophila* and *Vibrio Cholera* chitinases have been found responsible for promoting bacterial colonization of lungs and intestine, respectively [128]. It is conceivable that the presence of a chitin-binding motif on bacterial chitinases favors bacterial adherence to the surface of host epithelial cells under inflammatory conditions [129]. This has been confirmed for both CBP21 of *Serratia Marcescens* and ChiA of *AIEC* which interact with CHI3L1 to attach to intestinal epithelial cells [8].

The post-translational modifications of CHI3L1 presents an N-glycosylated protein with two molecules of GlcNAc at the 60th asparagine residue in human [130]. It is noteworthy how the extent of glycosylation of both host and microbiomes changes in the context of bacterial infection. The resulting glycome becomes an expression of highly complex glycosylated ligands which serve as receptors and primary sites of contact for bacteria [131]. In particular, N-linked surface glycoproteins expressed on host cells are likely to be a target for bacterial chitinases. For instance, *Salmonella Typhimurium* links to sugar compounds on apical host cells with high specificity, thus showing preference among the glycosylated moieties.

Moreover, alterations of the glycome occur before bacterial entry which proves them to be a direct consequence of host–microbial interaction [132]. Accordingly, it is possible to infer that N-glycosylation of host CHI3L1 is one of the critical steps to bacterial binding. Glycosylation of epithelial cells highly depends on the integrity of the sub- and supra-mucosal environment. Flaws in mucus glycosylation can yield a degraded mucus layer and less efficient segregation between host and intact bacteria [133,134] (Figure 3). In addition, glycosylated CHI3L1 plays a key role in host–microbial interactions since the mutation in CHI3L1 60th or 68th asparagine residue in human or mice, respectively, results in the reduction of bacterial adhesion to CECs [9] (Figure 3). 

Mucosal disruption is a typical finding in IBD cases as well as in the context of a systemic inflammatory response [135]. Specifically, it has been documented that CHI3L1 is overexpressed in the tissue upon bacterial colonization of severe burn injuries. Again, the interruption of the surface barrier promotes bacterial contact with the underlying epithelium, thus accounting for increased chitinase levels.

Following adhesion, the invasion of bacteria into the colonic epithelium is the end result of complex cellular mechanisms involving both sides of the equation. The release of pro-inflammatory cytokines, primarily TNF-α, IL-1β, and IL-6, fosters a comparable CHI3L1 expression on host cells. To this extent, it has been established that these proinflammatory factors, especially TNF-α, can induce the expression of CHI3L1 mRNA and late secretion of CHI3L1 protein. In turn, CHI3L1 can activate the several signaling pathways, including NF-κB and MAPK signaling cascades, which promote chronic inflammatory conditions. 

Another key step to bacterial penetration is the polarization of macrophages. The CHI3L1-driven M2 transition is part of a compensated anti-inflammatory response. However, in a dysregulated environment, the M2 presence hinders the pro-inflammatory defenses owing to poor antigenic properties. This leads to an equally poor bacterial clearance because the engulfed pathogens reside internally within the mucosa. Interestingly, bacteria, like Staphylococcus aureus, exploit this mechanism to evade immune recognition [136]. Others, such as adherent invasive *Escherichia coli* (AIEC), keep replicating within macrophages. These data indicate how commensal bacteria can both start and uphold enteric inflammation. 

### 3.2. Interactions between CHI3L1 and Bacterial Chitinase (ChiA) in Escherichia coli 

*E. coli* is one of the major representatives of commensal bacteria producing bacterial chitinase. Particularly, AIEC is normally present in the intestinal flora of healthy individuals but shows increased numbers in acute CD patients. This finding suggests that AIEC strains display pathogenicity in susceptible hosts via increased adhesion to host cells. The disrupted mucus layer, typical of IBD, and the CHI3L1 upregulation make enteric epithelial cells accessible to AIEC strains (Figure 3). Indeed, AIEC’s primary interaction consists in binding the host CHI3L1 via the bacterial chitinase, ChiA. Particularly, it has been recorded that ChiA overexpression occurs in AIEC strains rather than non-pathogenic *E. coli* strains, thus accounting for microbe-specific features [9,137].

Similarly to mammalian chitinases, the genotype of ChiA can influence the rate of invasion of *E. coli* into host CECs [9]. The presence of polymorphisms in the chitin-binding domains within ChiA allows clustering of *E. coli* strains according to their relative pathogenicity, which is measured in terms of adhesiveness to CECs [9].

Multiple factors compromise the host microenvironment and favor bacterial access to the intestinal epithelium. First of all, host macrophages release pro-inflammatory cytokines in conditions of chronic inflammation. It is noteworthy how inflammatory cytokines, in particular TNFα, foster a greater expression of CHI3L1 and, thus, a wider site of contact for bacterial chitinases [138]. 

The post-translational N-glycosylation of CHI3L1 is crucial for an efficient host–microbe interaction [9] (Figure 3). In addition, the expression of glycosylated moieties, namely CEACAM6, is necessary for AIEC adhesion. Similarly to CHI3L1, CEACAM6 serves as a binding receptor for the bacterial appendices and is expressed upon TNFα stimulation following AIEC infection [126]. These data confirm how commensal bacteria can sustain colonization by exploiting the modification of host cells by glycation [9,126].

### 3.3. Potential Role of CHI3L1 as an Inducer of Intestinal Dysbiosis 

The alteration of the enteric microbiota is associated with a wide variety of gastrointestinal diseases. Intestinal dysbiosis may present as the source, the result, and, most frequently, the sustainer of chronic inflammatory states [4,15]. The composition of gut microbiota is modulated by several factors, some of which are unmodifiable, such as the immune system, the enteric mucosa, and the microbiome. This finding is supported by the pathophysiology of IBD, which usually presents mucus disruption, immune dysregulation, and dysbiosis [4,15].

Any imbalance among the bacterial taxa can lead to reduced microbial diversity and predominance of pathogenic strains. These favor disease development and severity by impairing intestinal homeostasis and promoting immunosuppression and cancer cell growth. In this context, the host–microbial interaction plays a central role. It has been demonstrated that CHI3L1 enhances bacterial adhesion in susceptible hosts. Interestingly, it is possible that CHI3L1 preferentially engages pathogenic (e.g., *S. typhimurium)* and potentially pathogenic (e.g., AIEC) strains rather than non-pathogenic ones (e.g., DH5α) [9,13,139]. This mechanism would reinforce the extent of microbial penetration within CECs and further contribute to intestinal dysbiosis. In addition, the formation of a bacterial biofilm on the surface of the colonic epithelium is related to the pathogenic transition of certain bacterial strains. This finding suggests that the loss of mucosal protection and the increased intestinal permeability induce the intramucosal replication of intact bacteria that are normally excluded from colonic tissue. Altogether, these events contribute to shaping intestinal flora and affecting immune tolerance. 

Mice remain some of the best animal models for investigating changes in microbiota presentation. It has been shown that, following bacterial infection, chemically-induced colitis, or immune deficiency, mice enteric flora develops a lower number of total commensal bacterial as well as a reduced richness in resident strains with respect to normal controls [140]. These data underlie how different sources of inflammation can account for intestinal dysbiosis. 

IBD is one of the most representative cases of chronic intestinal dysbiosis. Despite the multifactorial nature of the disease, the alteration in the microbiota composition is rather relevant. Most of the bacterial phyla in a healthy intestinal flora are *Firmicutes*, *Bacteroidetes*, *Proteobacteria*, and *Actinobacteria*. In IBD patients, Bacteroidetes and Proteobacteria are more abundant, whereas *Firmicutes* are reduced [141]. Moreover, the microbial richness diminishes with evidence of predominant strains and clusters, such as *Enterobacteriaceae* and *Bilophila* for Proteobacteria and *Faecalibacterium prausnitzii* for Firmicutes. This background might lead to metabolic changes that affect the whole gut homeostasis. In addition, the amount of mucus-degrading bacteria, such as *Ruminococcus gnavus* and *Ruminococcus torque*, is significantly higher in IBD with respect to normal controls [142]. Thus, contributing to reduced mucus protection and increased epithelial exposure to commensal bacteria. 

Overall, the data above suggest the potential role of host CHI3L1 in shaping the intestinal biome and favoring the penetration of potentially pathogenic bacteria in normal flora under inflammatory conditions. This evidence suggests a prospective therapeutic target for the treatment of IBD by inhibiting CHI3L1 expression in the attempt to exclude the entry route of invasive species from the aftermath of intestinal dysbiosis.

## 4. Therapeutic Potentials of CHI3L1-Blockers/Inhibitors for Various Diseases 

### 4.1. Anti-CHI3L1 Antibody

Given the pivotal role of CHI3L1 in chronic inflammation, targeting this protein for therapeutic purposes has gathered significant interest. Choi et al. investigated the inhibitory potential of 2-({3-[2-(1-cyclohexen-1-yl)ethyl]-6,7-dimethoxy-4-oxo-3,4-dihydro-2-quinazolinyl}sulfanyl)-N-(4-ethylphenyl) butanamide (K284-6111) in an AD mouse model [102]. Their study revealed that K284-6111 binds to CHI3L1, thereby inhibiting its interaction with the receptor for RAGE (Receptor for Advanced Glycation End Products). This interaction led to the suppression of NF-κB activation and NF-κB-related neuro-inflammatory gene expressions in mice. Furthermore, K284-6111 was evaluated in a phthalic anhydride (5% PA)-induced atopic dermatitis animal model, where topical application attenuated dermatitis severity, epidermal hyperplasia, inflammatory cell infiltration, and release of inflammatory cytokines [143]. In addition to small molecules like K284-6111, humanized antibodies targeting CHI3L1 have been developed and tested for their efficacy.

CHI3L1 antibodies have earned considerable interest among researchers for their potential application in cancer therapy. In the context of chronic inflammation, CHI3L1 plays a crucial role in fostering tumor-associated inflammation and shaping the tumor microenvironment (Figure 2). Yang et al. developed polyclonal neutralizing anti-CHI3L1 antibodies (nCHI3L1 Abs) and assessed their efficacy in lung, pancreas, and colon cancer allograft models [143]. Their findings demonstrated that nCHI3L1 Abs effectively inhibited tumor growth and metastasis in orthotopic lung, pancreatic, and colon cancer models. Additionally, in vitro studies confirmed the ability of nCHI3L1 Abs to suppress the AKT, β-catenin, and NF-κB signaling pathways [143]. Furthermore, Yu et al. reported similar findings, wherein their anti-CHI3L1 antibody exhibited efficacy in reducing lung tumor growth and metastasis by inhibiting M2 polarization [144].

Moreover, CHI3L1 exerts a significant regulatory influence on immune checkpoints. In a melanoma lung metastasis mouse model, Ma et al. demonstrated that CHI3L1 upregulates the expression of programmed death-ligand 1 (PD-L1) on activated macrophages while concurrently suppressing the expression of Inducible T-cell Co-Stimulator (ICOS), ICOS Ligand, and CD28 on T cells and antigen-presenting cells [145,146]. Their anti-CHI3L1 antibody (so-called FRG), PD-1 antibody, and CTLA-4 antibodies exhibited substantial anti-tumor effects and displayed additive responses in metastasis models. Intriguingly, in vitro studies confirmed synergistic cytotoxic effects on tumor cells, while significantly enhanced anti-tumor responses were observed in in vivo tumor models treated with bispecific antibodies targeting both FRG and PD-1. Similar effects were confirmed with bispecific antibodies targeting both FRG and CTLA-4 [146,147]. 

In recent clinical practice, immune checkpoint inhibitors (ICIs) have become commonplace. Nevertheless, with the utilization of ICIs, a notable 43% of patients report experiencing immune-related adverse events (irAEs) [147]. Considering that the most prominent complications associated with ICIs encompass chronic immune-related adverse events like dermatitis, hepatitis, arthritis, and colitis, and recognizing CHI3L1 as a significant immune checkpoint modulator, the concurrent targeting of CHI3L1 using bispecific antibodies may represent a prospective solution [148,149].

### 4.2. Methylxanthine Derivatives including Caffeine as a Pan-Chitinase Inhibitor

Methylxanthines, including caffeine, pentoxifylline, theophylline, and allosamidin, are a group of alkaloids that are derived from the purine-based xanthine [150]. Interestingly, it has been demonstrated through the use of drug screening tools that several methylxanthine derivatives potentially work as pan-chitinase inhibitors [151]. Allosamidin, a chitinase inhibitor produced by *Streptomyces*, showed a higher affinity against fungal chitinase as compared to caffeine, pentoxifylline, and theophylline [150]. Based on X-ray diffraction analysis showed that all the three methylxanthine derivatives listed above have a common binding position for family 18 chitinases, just like allosamidin and working as a pan-chitinase inhibitor. 

Based on the characteristic feature of methylxanthine derivatives as pan-chitinase inhibitors, our group compared the influence of CHI3L1 m RNA expression levels in SW480, a human colon epithelial cells, after treatment with caffeine, pentoxifylline, theophylline [150]. As a result, all three methylxanthine derivatives directly downregulated the CHI3L1 mRNA expression levels in SW480 cells in a dose-dependent manner [151]. Since CHI3L1 is induced on epithelial cells and macrophages during inflammatory conditions as well as inflammation-associated cancer states by activating several important signaling pathways, including AKT and β-catenin, thus methylxanthine derivatives have potential anti-inflammatory and anti-cancer effects through the inhibition of CHI3L1. In fact, oral caffeine administration at the concentration of 2.5 mM efficiently prevents the onset of a murine model of acute colitis by DSS (dextran sulfate sodium) [152]. The caffeine-mediated anti-inflammatory effect was exerted by suppressing CHI3L1 and AMCase but not chitinase -1 as determined by quantitative PCR of colonic tissue after induction of DSS-acute colitis [152]. However, a paradoxical effect of caffeine was also identified: low-dose (0.17 mM) caffeine with 10% sucrose (but not fructose) caused apparent carcinogenic change in CECs in a murine model of chronic colitis [153]. Therefore, we need to carefully determine the influence of methylxanthine derivatives as a potential anti-inflammatory therapeutic strategy.

### 4.3. Chitin Microparticles and Chito-Oligosaccharides

Chitin is a polymer of GlcNAc, which is produced by fungi, crustaceans, insects, etc. [154,155]. Chitin is the second most abundant polysaccharide in nature, next to cellulose [154]. In addition, chitin is a valuable biological resource that is estimated to be synthesized on earth in an amount of 100 billion tons per year. However, chitin is difficult to utilize because it cannot be dissolved in ordinary solvents and can only be broken down by enzymatic active true chitinases. Interestingly, chitin has different biological effects depending on its size: Large chitin particles (diameter > 100 μm) are non-functional, medium chitin particles (40–70 μm) are pro-inflammatory function, and chitin microparticles (CMPs: 1–10 μm) are believed to be anti-inflammatory as well as regulatory functions by stimulating IL-10 production [156,157,158,159]. Both host CHI3L1 and bacterial chitin-binding proteins (CBPs) are characterized by their ability to bind to chitin [7]. These findings encouraged us to propose a “Trimetric“ Interaction model in the interaction of CHI3L1 and CBP is mediated by exogenous/endogenous chitin or chitin-like oligosaccharides, forming a CHI3L1/chitin/CBP trimeric complex (Figure 4). However, our published data [9] now suggest that glycosylated (60th asparagine in humans and 68th asparagine in mice) CHI3L1 can directly interact with CBP, promoting us to propose an alternative mechanism called “dimeric” interaction theory (Figure 3 and Figure 4). In addition, we recently found AIEC LF82 strain attachment on CECs was abolished when these cells were treated with N-glycosylation inhibitor (tunicamycin) or engineered to overexpress mutant CHI3L1 lacking 68th asparagine (N68P mutant), a site of N-glycosylation for CHI3L1 [9] (Figure 3). These results raise a possibility that N-glycosylated CHI3L1 is essential for the interaction between CECs and potentially pathogenic bacteria under inflammatory conditions.

Furthermore, CMPs inhibit, rather than enhance, the interaction of CECs and pathogenic bacteria and efficiently modulate intestinal inflammation in vivo [159]. As mentioned previously, chitin particles play different biological roles depending on their size. Since chitin is unstable, it is difficult to generate small chitin of the same size, which has hampered investigators’ abilities to dissect the biological role of chitin more closely and accurately, in particular in vivo. To overcome this problem, water-soluble and equal-sized chito-oligosaccharides (CHOS) nano-particles (1–10 nm in size) must be more useful [160] for in vivo studies as therapeutic strategy for inflammatory disorders including IBD and COPD since CHOS is the very end-product of chitin and does not need to be dissected anymore. 

## 5. Conclusions

As a result of many research reports to date, it has become clear that the expression of CHI3L1 on various types of cells is deeply involved in the process of chronic inflammation and carcinogenesis. Therefore, inhibiting CHI3L1 expression is expected to be a new prevention and treatment strategy for chronic inflammation as well as inflammation-associated cancer. CHI3L1 expression is positively associated with increased angiogenesis and metastasis in highly malignant tumors such as colorectal cancer, lung cancer, and glioblastoma. It is expected that it could also contribute to the treatment of not only inflammation but also cancer by inhibiting CHI3L1 expression in multiple ways including anti-CHI3L1 specific antibodies, methylxanthine derivatives, and chitin microparticles.

## Figures and Tables

**Figure 1 cells-13-00678-f001:**
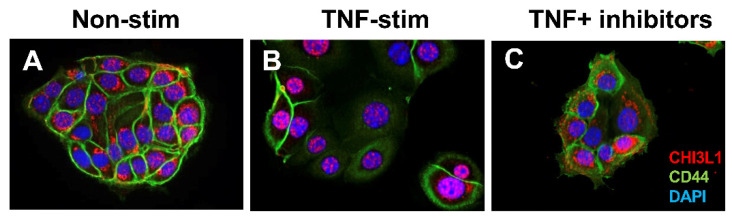
Nuclear translocation of CHI3L1 after TNFα stimulation in colonic epithelial cells (CECs): Human CEC line HCT116 cells have been cultured for 48 h on 3-dimensional Matri-gel without stimulation (**A**), with 50 ng/mL TNFα stimulation (**B**), or with 50 ng/mL TNFα stimulation with chitinase inhibitors (mixture of 2.5 mM caffeine and 25 mM pentoxifylline) (**C**). The cells were stained with anti-CHI3L1 antibody (shown as red) and anti-CD44 (shown as green) followed by CF594 anti-rabbit IgG and CF488 anti-mouse IgG, respectively. The nucleus is stained by DAPI (shown as blue) and analyzed by fluorescence microscope BZ-X800 (Keyence, Osaka, Japan) at 100× objective lens with oil.

**Figure 2 cells-13-00678-f002:**
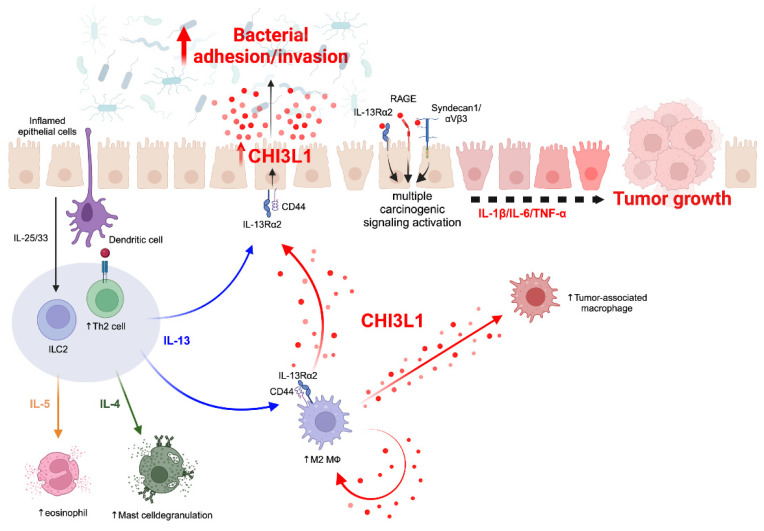
Conceptual representation of the putative effects of CHI3L1 in chronic inflammation within epithelial cells: Bacterial binding to epithelial cells triggers the production of pro-inflammatory cytokines. This inflammatory cascade results in the upregulation of CHI3L1 expression in epithelial cells. Additionally, innate lymphoid cells type 2 (ILC2) and Th2 cells initiate a Th2 inflammation in response to epithelial cell inflammation. Th2 inflammation further promotes the polarization of M0 macrophages towards the M2 phenotype. IL-13 plays a crucial role in inducing CHI3L1 expression in both epithelial cells and M2 macrophages. Elevated CHI3L1 expression in epithelial cells enhances bacterial adhesion and sustains chronic inflammation. Soluble CHI3L1 binds to several receptors, including IL-13Rα2, RAGE, and syndecan I on epithelial cells and macrophages. Moreover, chronic inflammation mediated by CHI3L1 leads to tumorigenesis, contributing to the establishment of a tumor-promoting microenvironment. Created with BioRender.com (accessed on 19 February 2024).

**Figure 3 cells-13-00678-f003:**
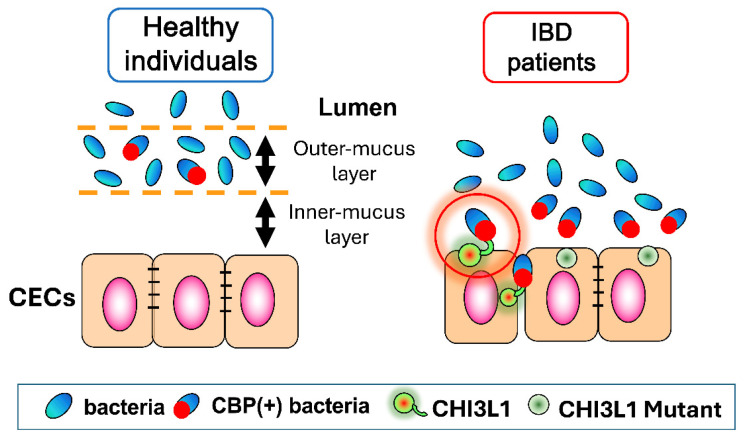
Intestinal microflora and host colonic epithelial cells’ (CEC) interactions in healthy individuals versus inflammatory bowel disease (IBD) patients: In healthy individuals, CEC is protected by two (outer- and inner-) mucus layers, and intestinal bacteria cannot access CECs. In contrast, in IBD patients, disruption of these mucus layers causes the chitin-binding protein of potentially pathogenic bacteria to bind to glycosylated CHI3L1 expressed on CEC, allowing these bacteria to adhere and invade. Non-glycosylated mutant form of CHI3L1 shows less bacterial binding in vitro.

**Figure 4 cells-13-00678-f004:**
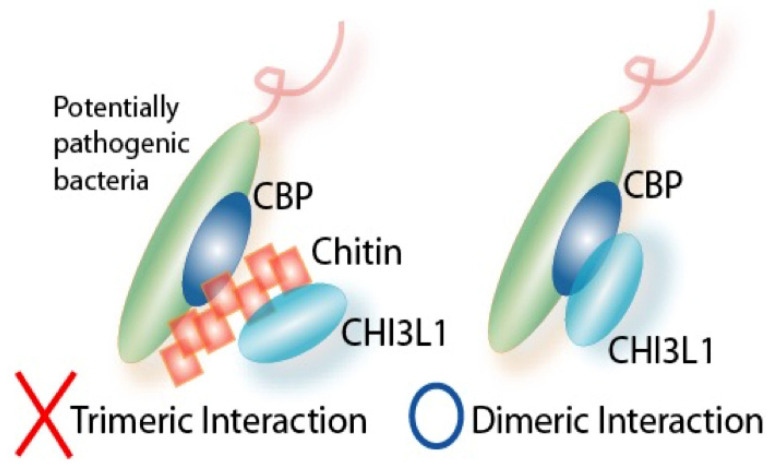
Schematic representation on putative trimeric- and dimeric-interaction theories: Our current hypothesis is based on a dimeric interaction theory (a direct interaction of host N-glycosylated CHI3L1 protein and bacterial chitin-binding protein) but not a trimeric interaction theory (chitin or chitin-like oligosaccharides link CHI3L1 protein and chitin-binding protein together make those trimeric complex).

**Table 1 cells-13-00678-t001:** Increased levels of CHI3L1 under inflammatory conditions in humans referred within the past five years. Classification: ①Autoimmune, ② Infectious, ③ Genetic, and ④ Degenerative inflammation.

Disease	Samples	General Overview	Refs.
Intestine (IBD) ①②③	Serum (<9 folds ↑in adults) CECs (<14 folds ↑in adults), Stool (<288 fold ↑ in adults), Mφ, Peripheral MDMs	Increased CHI3L1 expression levels in CECs enhance the interaction with intestinal microbiota. Anti-TNF agents can restrict replication of pathobionts by modulating CHI3L1 in CD. CHI3L1 plays the interactions between innate and adaptive immune response in IBD.	[23,24,25,26]
Neuron (MS) ①③④	CSF (<9 folds ↑in adults), Serum (<2 folds ↑in adults), Tissue (<2 folds ↑in adults)	CHI3L1 in CSF is a marker for MS progression. Patients with MS had higher CSF level of CHI3L1 than age- and sex- matched healthy control. CHI3L1 expression levels was strongly associated with microglial activation in the white matter. Serum CHI3L1 was increased in group with severe disease activity. The CNF sTNF-R1-associated pattern including CHI3L1 was related to altered T-and B-cell signaling. CHI3L1 may be associated with low-grade non-lymphocytic inflammation and active neurodegeneration. Increased CSF CHI3L1 levels were obtained in untreated MS patients compared to symptomatic controls. CSF CHI3L1 levels were increased in LS-OCMB absent patients as compared to control.	[27,28,29,30,31,32,33,34,35]
Neuron (AD) ①④	CSF (<5 folds in adults), Tissue(<10 folds ↑in adults)	CHI3L1 is one of the five loci showed genome-wide significant association with CSF profile. CHI3L1 functions as a signaling molecule mediating distinct neuroinflammatory responses in brain cells. CHI3L1 expression level is positively associated with postsynaptic damage and microglial activation. White matter CHI3L1 inflammatory response is associated with cognitive impairment early in the onset. CSF CHI3L1 levels were tightly related to CSF tau and p-tau levels in the MCI group. CHI3L1 may help in identifying early brain changes during the onset of AD. Increased CSF CHI3L1 was noted only at the MCI stage.	[36,37,38,39,40,41,42,43]
Bronchus/Lung (Asthma, COPD) ①③	Serum(<2 folds increased), BECs	Increased serum CHI3L1 in asthma patients could be used as an emerging indicator for the disease. A positive association between the assessed cytokines and CHI3L1 in moDCs was observed in asthma and COPD. miR-149-5P directly regulates CHI3L1 in context of TLR-mediated airway epithelial cell inflammation. Serum CHI3L1 was an independent biomarker of negative responses to anti-asthma regimens.	[44,45,46,47,48]
Bronchus/Lung (COVID-19) ①	Serum (<42 folds increased), Tissue	Serum CHI3L1 levels in patients with severe disease were significantly higher than the other three groups. Slightly high plasma CHI3L1 in COVID-19 patients with a more unfavorable outcome than non-ICU survivors. CHI3L1 may serve as a highly sensitive prognostic marker for COVID-19. CHI3L1 is upregulated in a tissue-specific marker in the liver after SARS-CoV2 infection.	[49,50,51,52]
Lung (CF) ①②③④	Stool (<6 folds ↑in children), Serum	Children with CF had higher fecal CHI3L1 than healthy control. Plasma CHI3L1 levels of pediatric and adult CF at all periods were significantly higher than control.	[53,54]
Liver (LC, CHC) ①②③④	Serum, Tissue	Serum CHI3L1 levels can be used to monitor changes in fibrosis in CHC patients. CHI3L1 is one of the four independent NASH-associated biomarkers. CHI3L1 protects the liver function from APAP injury by inhibiting the secretion of inflammatory factors.	[55,56,57]
Kidney (ESKD, DKD) ①②③④	Serum, Urine (<7 folds ↑in adults)	Serum CHI3L1 was increased in deceased males with ESKD as compared to those of females. Urine CHI3L1 level was associated with greater eGFR decline. Plasma CHI3L1 was associated with a greater risk of progression of diabetic kidney disease than control. Plasma CHI3L1 was not independently associated with progression of DKD. Urine CHI3L1 was associated with higher risk of the kidney composite outcome.	[58,59,60,61,62]
Heart (CHD, AS) ③④	Serum (<5 folds increased)	Plasma CHI3L1 levels might be useful to identify a risk of cardiovascular death in patients with chronic CHD. CHI3L1 is one of the 11 cardiovascular proteins with causal evidence of involvement in human disease. Plasma CHI3L1 levels were elevated in aortic AS and associated with mortality.	[63,64,65]
Pancreas (DM) ①③④	Serum (<134 folds increased)	CHI3L1 is a useful marker of intestinal permeability and inflammation of Type 2 Diabetes Mellitus. Vitamin D might contribute to reducing diabetic complications via modulating CHI3L1 and MCP-1 signaling pathways.	[66,67]
Oral Cavity ②③	Tissue, Saliva (<8 folds ↑ in adults)	Increased CHI3L1 expression in the intraoral tissue from inflammatory lesions. Advanced dental caries with pulp exposure are positively associated with the increasing levels of CHI3L1 in saliva.	[68,69]
Nose (Rhinitis) ①②③	Serum	Serum CHI3L1 levels were significantly decreased in patients with allergic rhinitis than control group.	[70]
Joint (RA) ①③④	PBMCs	CHI3L1 gene/protein expression levels were suppressed in PBMCs from RA patients after anti-TNF treatment.	[71]

Abbreviations: AD, Alzheimer’s disease; AIEC, adherent invasive Escherichia coli; AS, aortic stenosis; BECs, bronchial epithelial cells; APAPS, Acetaminophen; CECs, colonic epithelial cells; CF, cystic fibrosis; CHC, chronic hepatitis C; CHD, coronary heart disease; CHI3L1, chitinase 3-like 1; CKD, chronic kidney disease; CSF, cerebrospinal fluid protein; COPD, chronic obstructive pulmonary disease; COVID-19, Corona virus infectious disease, emerging in 2019; DKD, diabetic kidney disease; DM, diabetes mellitus; eGFR, estimated glomerular filtration rate; ESKD, end-stage kidney disease; GWAS, genome-wide association studies; IBD, inflammatory bowel disease; ICU, intensive care unit; LC, liver cirrhosis; MCI, mild cognitive impairment; LS-OCMB, lipid-specific oligoclonal IgM bands; MDMs, monocyte derived-macrophages; moDCs, monocyte derived dendritic cells; MS, multiple sclerosis; Mφ, macrophages; NASH, non-alcoholic steatohepatitis; RA, rheumatoid arthritis; SARS-CoV2, Severe acute respiratory syndrome coronavirus 2; TLR, Toll-like receptor; TNF, tumor necrosis factor, TNF-R1, TNF-receptor type I. Arrow in this table represents the increased amount of each sample.

**Table 2 cells-13-00678-t002:** CHI3L1-associated chronic inflammation in animal models.

Disease	Model	Features	Refs.
Eosinophilic Chronic Rhinosinusitis	CC10 WT/KO mice with OVA sensitization	mRNA and protein CHI3L1 levels were high in the allergic ECRS model. CC10 regulates ECRS by attenuating CHI3L1 expression.	[100]
Alzheimer’s disease	5× FAD mouse model Aβ_1–42_-induced AD mouse model	CHI3L1 in CSF was elevated with disease progression. CHI3L1 induces neurotoxicity and suppresses neural electrical activities. CHI3L1 KO decreases Aβ accumulation. CHI3L1 inhibition might suppress amyloidogenesis and neuroinflammation via inhibition of NF-κB.	[101,102]
Parkinson’s disease	LPS-induced PD rat model	Increased CHI3L1 in the brain tissue and CSF were noted PD rats group.	[103]
Multiple sclerosis	EAE-PLP mouse model	High CHI3L1 expression was noted in oligodendrocytes in the EAE model.	[104]
Atopic dermatitis	Filaggrin mutated mice with OVA sensitization	BRP-39 protein expressions in serum and skin were increased in the AD mouse model. BRP-39 KO diminished Th2 inflammatory responses.	[105]
Asthma	BRP-39 WT/KO mice with OVA sensitization	*Chi3l1* mRNA and protein were upregulated during the aeroallergen-induced Th2 inflammation and IL-13 effector responses. CHI3L1 is critical for IL-13 to induce tissue inflammation and fibrosis.	[106]
COPD	BRP-39 WT/KO mice with cigarette smoke	*Chi3l1* mRNA and protein were upregulated after 10 months of cigarette smoke. BRP-39 KO showed significantly reduced cigarette smoke-induced BAL & tissue inflammation. BRP-39 KO enhanced cigarette smoke-induced epithelial cell apoptosis and alveolar destruction.	[107]
Hermansky–Pudlak syndrome	*Hps1* mutation mice with Bleomycin	*Hps1* mutation model showed exaggerated fibroproliferative response by CHI3L1 binding to CRTH2 The inability of CHI3L1 to bind to IL-13Rα2 leads to severe injury and apoptosis.	[108]
Atherosclerosis	ApoE^−/−^ mice with high-fat diet	CHI3L1 KD showed larger size and less stable atherosclerosis plaques.	[109,110]
Liver sepsis	LPS-induced mouse model	CHI3L1 KO mice showed reduced M2 polarization markers but no change in the WT group. CHI3L1 KO mice demonstrated a higher survival rate than the WT group.	[111]
Alcoholic liver injury	CHI3L1 WT/KO mice with the Lieber-DeCarli ethanol liquid diet	mRNA levels of Acetyl-CoA carboxylase, fatty acid synthase, and stearoyl-CoA desaturase-1 were increased by ethanol, but they were suppressed in the CHI3L1 KO mice. The upregulated oxidative stress and pro-inflammatory cytokines were attenuated in the CHI3L1 KO mice.	[112]
NASH	CHI3L1 WT/KO mice (Cre^Lyz^) with choline-deficient high-fat diet	CHI3L1 protein expression was increased in the NASH model. The Myeloid-specific CHI3L1 KO NASH model demonstrated a significantly reduced accumulation of pro-inflammatory macrophages and neutrophils compared with the WT mice.	[113]
Chronic liver injury	CCl_4_ i.p. injection rat model	mRNA and protein CHI3L1 expression showed a sustained increase for the chronic term. CD14^+^ cells, such as Kupffer cells, were the resource of the CHI3L1.	[114]
Obesity	CHI3L1 WT/KO mice with HFD	*Chi3l1* mRNA was elevated in the HFD group. CHI3L1 KO mice showed a reduced accumulation of white adipose tissue. mRNA of TNF-α, IL-6, and IL-10 were significantly lower in the white adipose tissue from the CHI3L1 KO mice.	[92]
CKD	Ischemia/reperfusion injury mouse model with microaneurysm clip	*Ccl2* and *Chi3l1* mRNA levels were higher in infiltrating macrophages and neutrophils, respectively. They also correlate with atrophy and renal fibrosis.	[59]
Osteoarthritis	Osteoarthritis rat model with ACLT	Immunohistochemical analysis showed that CHI3L1 staining was prominent in osteoarthritic cartilage, especially in the superficial areas of the cartilage.	[115]
Rheumatic arthritis	RA mouse model with HC gp-39 (CHI3L1) i.p. injection	HC gp-39 induced higher clinical scores of arthritis in a dose-dependent manner. The HC gp-39 injected mice showed infiltration of mononuclear cells and synovial fibroblast proliferation in their ankles.	[116]
IBD	CHI3L1 WT/KO with AOM/DSS CHI3L1 WT/KO with *S.typhimurium*/AIEC inoculation	During the chronic phase of colitis, CHI3L1 expression was significantly elevated in both serum and stool. CHI3L1 binds to RAGE and activates STAT3 and β-catenin, which creates favorable conditions for neoplastic changes. CHI3L1 is essential for both *S. typhimurium* and AIEC LF82 induced to induce severe intestinal inflammation. CHI3L1 and IL-6 synergistically activate the colonic epithelial STAT3 signaling pathway.	[74,79]

Abbreviations: Aβ, amyloid-beta; ACLT, anterior cruciate ligament transection; AD, Alzheimer’s disease; AIEC, adherent invasive *Escherichia coli*; ApoE, apolipoprotein E; AOM, azoxymethane; BAL, bronchoalveolar lavage; BRP-39, breast regression protein-39; CC10, clara cell 10kD; *Ccl2*, C-C motif chemokine ligand 2; CCl_4_, carbon tetrachloride; CHI3L1, chitinase 3-like 1; CKD, chronic kidney disease; CRTH2, chemoattractant receptor homologous molecule on T helper type 2 cells; COPD, chronic obstructive pulmonary disease; CSF, cerebrospinal fluid; DSS, dextran sulphate sodium; EAE, experimental autoimmune encephalomyelitis; ECRS, eosinophilic chronic rhinosinusitis; HC gp-39, human cartilage glycoprotein-39; HFD, high-fat diet; HPS, Hermansky-Pudlak syndrome; IL-13Rα2, interleukin-13 receptor alpha 2; i.p., intraperitoneal; KD, knock down; LPS, lipopolysaccharides; MS, multiple sclerosis; NASH, non-alcoholic steatohepatitis; OVA, ovalbumin; PD, Parkinson’s disease; PLP, proteolipid protein; RA, rheumatoid arthritis; RAGE, receptor for advanced glycation end products; STAT3, signal transducer and activator of transcription 3; *S. typhimurium*, Salmonella typhimurium.

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
