# Peer review of "Recently Updated Role of Chitinase 3-like 1 on Various Cell Types as a Major Influencer of Chronic Inflammation"

_cells, 2024, doi:10.3390/cells13080678_

Round 1
Reviewer 1 Report
Comments and Suggestions for Authors
The review by Mizoguchi et al regarding the association and potential roles of CHI3L1 in disease is very informative and is generally well written and organized. The amount of information compiled and presented is impressive. I have only minor comments / suggestions:
- The beginning (lines 31-43) and end (lines 72-81) is somewhat redundant
- Combine the first 3 paragraphs in section 3.1
Comments on the Quality of English Languagegood quality in general, minor edits needed
Author Response
1) Beginning (lines 31-43) and end (lines 72-81) is somewhat redundant.
Answer)We deeply appreciate this important advice and have removed the two suggested points (the beginning and the end parts) from the revised manuscript.
2)Combine the first 3 paragraphs in section 3.1.
Answer)As suggested, we have combined the first 3 paragraphs in section 3.1 in the revised manuscript.
Reviewer 2 Report
Comments and Suggestions for Authors
This review addresses the importance of mammalian chitinase 3-like 1 (CH13L1). CH13L1 is induced in a variety of pathological inflammatory conditions and there is evidence that it may play a key role in the development of these disorders. Evidence for this is provided in the review.Overall the review is well organised and comprehensively covers relevant aspects of the role of CH13L1 in inflammatory disease ,expression in different cell types ,how it interacts with different microbial species in the inflammatory response as well as being a target for therapeutic intervention in inflammatory disease. It comprehensively covers its possible role in two tables and is well illustrated to represent its likely role in the inflammatory response.
Specific points
1. Reference is made to CH13L1 elevation/significant increases in many different processes. It would help to state levels in the more important examples to get greater idea of the extent of elevation. Is it just over 1.00 or 10 f0ld?
2. Should be made clearer whether CH13L1 is regulated primarily by cytokine pathways or whether is does to regulating.
3. Is CH13L1 mediating pathogenic events or associated with these ? May be overstating its role.
4. What is the evidence that it mediates ABeta-induced tau phosphorylation and neurological injury?
5. Therapeutic potential overstated by inhibition e.g. treatment of IBD ,progression of AD,obesity related asthma and COPD. A wonder drug!!
Comments on the Quality of English LanguageJust some fine points in Tables etc
Author Response
1. Reference is made to CH13L1 elevation/significant increases in many different processes. It would help to state levels in the more important examples to get a greater idea of the extent of elevation. Is it just over 1.00 or 10 fold?
Answer)Thank you very much for your helpful comments. We revised the references in Table 1 according to the reviewer’s opinion and included the extent of elevation as mentioned in each cited article.
2. Should be made clearer whether CH13L1 is regulated primarily by cytokine pathways or whether it does to regulating.
Answer)Expression of CHI3L1 is mainly regulated by proinflammatory cytokines including TNFa, IL-1b and IL-6, but it is not certain that CHI3L1 is actually regulating the pro-inflammatory cytokine pathway(s) as a negative feedback loop, so we have toned down the description in the revised manuscript.
3. Is CH13L1 mediating pathogenic events or associated with these ? May be overstating its role.
Answer)Soluble CHI3L1 is binding with several receptors including IL-13Ra2, Syndecan 1/aVb3, and RAGE, so it is likely to mediate pathogenic events directly. We have mentioned this possibility in the revised manuscript.
4. What is the evidence that it mediates Abeta-induced tau phosphorylation and neurological injury?
Answer)As the Reviewer pointed out, there is no direct evidence that CHI3L1 directly mediates Amyloid beta-induced tau phosphorylation and neurological injury, so we have deleted the sentence in the revised manuscript.
5. Therapeutic potential overstated by inhibition e.g. treatment of IBD, progression of AD, obesity related asthma and COPD. A wonderful drug!!
Answer)We agree that we overstated the therapeutic effects of CHI3L1 in the previous version of the manuscript, so we toned down the effects in the revised manuscript.
Reviewer 3 Report
Comments and Suggestions for Authors
The topic of this review article aims to investigate the expression and function of CHI3L1 in inflammatory conditions and specific cells.
Despite the interesting choice of topic, the review article is extremely confusing, and the logical arc in which the authors present the topic is not clear.
In the introduction, they describe IBD, then discuss the role of the microbiome, then provide background information on CHI3L1, and then again on colonic epithelial cells. Even here, it is not clear how the chapters relate to each other.
Then, in Chapter 2, the role of CHI3L1 is presented, without any logical consideration, in a table listing several inflammatory conditions. They do not distinguish between autoimmune, infectious, genetic, and degenerative inflammation; they include vital pregnancy; and in the case of some pathologies, they overuse terminology (e.g., oral cavity disease, kidney disease, etc.).
Chapter 2.1 then uses some unknown logic to select the pathologies listed in Table 1. There is no immunopathological logic in this list either.
In chapter 2.2, the role of CHI3L1 in various cells is suddenly presented.
Here are synovial and cartilage cells, then sarcoma and cancer cells, followed by colon epithelial cells, colon adenocarcinoma cell lines, then glioblastoma cells and TAMs. What is the logic behind this seemingly random list? It is not at all clear.
Next, the CHI3L1-microbial component-colon epithelium interactions are presented. Here, they focus mainly on IBD and immune cells in the colon.
Then, in Chapter 3, the relationship between CHI3L1 and E. coli bacterial chitinase is presented, followed by a discussion of the role of CHI3L1 in dysbiosis.
Then, the role of CHI3L1 is presented through examples of animal models of inflammation. It is completely illogical why these randomly selected inflammatory animal models are listed after IBD. IBD is followed by neuroinflammatory diseases, metabolic disorders, then degenerative diseases, and then autoimmune diseases. But what was the authors' aim with this chapter? How does it relate to the previous ones?
The human CHI3L1 aspects will then continue with IBD after the presentation of asthma and HPS. Tumors are then discussed. What do these pathologies have in common? Why do the authors mention them?
The summary of therapeutic options targeting CHI3L1 is more or less correct.
Overall, the article seems to be a rather confused selection of random topics without any logic.
Although the topic is interesting, the article has serious shortcomings in terms of content and scientific logic, which make it unsuitable for publication in Cells.
Comments on the Quality of English LanguageMinor English language polishing is necessary.
Author Response
1. The topic of this review article aims to investigate the expression and function of CHI3L1 in inflammatory conditions and specific cells. Despite the interesting choice of topic, the review article is extremely confusing, and the logical arc in which the authors present the topic is not clear.
Answer)Thanks very much for the critical opinion. We are happy to know that our topic is interesting. We agree that the original manuscript is somewhat confusing and lacks clear logic. We revised this part in the revised manuscript. Hopefully, the revised manuscript is now acceptable.
2. In the introduction, they describe IBD, then discuss the role of the microbiome, then provide background information on CHI3L1, and then again on colonic epithelial cells. Even here, it is not clear how the chapters relate to each other.
Answer)As the reviewer pointed out, the order of presentation is quite confusing. Therefore, we moved the original Chapter 4 to Chapter 2. Hopefully, the revised manuscript provides a much better orientation for the topic.
3. Then, in Chapter 2, the role of CHI3L1 is presented, without any logical consideration, in a table listing several inflammatory conditions. They do not distinguish between autoimmune, infectious, genetic, and degenerative inflammation; they include vital pregnancy; and in the case of some pathologies, they overuse terminology (e.g., oral cavity disease, kidney disease, etc.).
Chapter 2.1 then uses some unknown logic to select the pathologies listed in Table 1. There is no immunopathological logic in this list either.
Answer) We greatly appreciate the Reviewer’s comments for Chapter 2. As the reviewer pointed out, we did not distinguish between autoimmune, infectious, genetic, and degenerative inflammation in the previous version. In the revised manuscript, we mentioned the four disease categories in each disorder. In addition, we removed the pregnancy and allergy from Table 1 as they are not apparent inflammatory disorders mediated by CHI3L1. We also have modified the category of oral cavity disease/ kidney disease according to the reviewer’s suggestion.
4. In chapter 2.2, the role of CHI3L1 in various cells is suddenly presented. Here are synovial and cartilage cells, then sarcoma and cancer cells, followed by colon epithelial cells, colon adenocarcinoma cell lines, then glioblastoma cells and TAMs. What is the logic behind this seemingly random list? It is not at all clear.
Answer)We agree with the reviewer’s opinion. In the revised manuscript, we arranged the order of orientation properly with logic. We have rearranged the order from inflammation-related cells to cancer-related cells.
5. Next, the CHI3L1-microbial component-colon epithelium interactions are presented. Here, they focus mainly on IBD and immune cells in the colon. Then, in Chapter 3, the relationship between CHI3L1 and E. coli bacterial chitinase is presented, followed by a discussion of the role of CHI3L1 in dysbiosis.Then, the role of CHI3L1 is presented through examples of animal models of inflammation. It is completely illogical why these randomly selected inflammatory animal models are listed after IBD. IBD is followed by neuroinflammatory diseases, metabolic disorders, then degenerative diseases, and then autoimmune diseases. But what was the authors' aim with this chapter? How does it relate to the previous ones?
Answer)Thank you very much for pointing out the weak points in the previous version of the manuscript. We completely agree with the Reviewer’s opinion. In the revised manuscript, “animal models of chronic inflammation” is listed in Chapter 2.3, followed by “CHI3L1-mediated host-microbial interactions” in Chapter 3.
In the revised Chapters 2 and 3, we would like to inform readers about the wide distributions of CHI3L1 in human general inflammatory conditions (in Table 1), CHI3L1-expressing cell types, CHI3L1 in chronic inflammatory animal models (in Table 2), and CHI3L1 in chronic inflammation in humans. Hopefully, the revised manuscript has satisfactorily addressed Reviewer 3’s concern.
6. The human CHI3L1 aspects will then continue with IBD after the presentation of asthma and HPS. Tumors are then discussed. What do these pathologies have in common? Why do the authors mention them? The summary of therapeutic options targeting CHI3L1 is more or less correct .Overall, the article seems to be a rather confused selection of random topics without any logics?
Answer)CHI3L1 is highly associated with chronic inflammation-associated tumorigenesis as we mentioned in some of our group’s publications in the new Ref. #74, 80, 95, 96, and 155. This concept, in particular colitis-associated cancer, is commonly accepted in the field of Gastroenterology, and our group is the pioneer in this topic. In addition, to prevent inflammation-associated early dysplasia/cancer, therapeutic strategies to settle the inflammation are so important. For this reason, we still included some inflammation-associated carcinogenic disorders in the revised manuscript. Hopefully, the revised manuscript is not so confusing for Reviewer 3. We carefully selected topics in the manuscript based on correct orientation and order. Therapeutic options of anti-CHI3L1 Ab, methylxanthine derivatives, and chitin, and the descriptions in the revised manuscript have been written based on published papers with reliable sources.
Round 2
Reviewer 3 Report
Comments and Suggestions for Authors
The revised version of the manuscript now much better than the original submission was.
The authors took into consideration the suggestions to improve their manuscirpt, which is now acceptable for publication.